# Mini-Review of Intra-Stark X-ray Spectroscopy of Relativistic Laser–Plasma Interactions

**Elisabeth Dalimier** [1,*], **Tatiana A. Pikuz** [2,3] **and Paulo Angelo** [1]

[1]  LULI—Sorbonne Université-Campus Pierre et Marie Curie, CNRS, Ecole Polytechnique, CEA: Université Paris-Saclay, CEDEX 05, F-75252 Paris, France; paulo.angelo@upmc.fr

[2]  Joint Institute for High Temperatures, Russian Academy of Sciences, 125412 Moscow, Russia; pikuz.tatiana@gmail.com

[3]  Open and Trans-Disciplinary Research Initiatives, Osaka University, 2-1, Yamadaoka, Suita, Osaka 565-0871, Japan

*  Correspondence: elisabeth.dalimier@upmc.fr

**Abstract:** Intra-Stark spectroscopy (ISS) is the spectroscopy within the quasi-static Stark profile of a spectral line. The present paper reviews the X-ray ISS-based studies recently advanced for the diagnostics of the relativistic laser–plasma interactions. By improving experiments performed on the Vulcan Petawatt (PW) laser facility at the Rutherford Appleton Laboratory (RAL), the simultaneous production of the Langmuir waves and of the ion acoustic turbulence at the surface of the relativistic critical density gave the first probe by ISS of the parametric decay instability (PDI) predicted by PIC simulations. The reliable reproducibility of the experimental signatures of PDI—i.e., the Langmuir-wave-induced dips—allowed measurements of the fields of the Langmuir and ion acoustic waves. The parallel theoretical study based on a rigorous condition of the dynamic resonance depending on the relative values of the ion acoustic and the Langmuir fields could explain the disappearance of the Langmuir dips as the Langmuir wave field increases. The ISS used for the diagnostic of the PDI process in relativistic laser–plasma interactions has reinforced the reliability of the spectral line shape while allowing for all broadening mechanisms. The results can be used for a better understanding of intense laser–plasma interactions and for laboratory modelling of physical processes in astrophysical objects.

**Keywords:** intra-Stark spectroscopy; relativistic laser–plasma interaction; parametric decay instability; X-ray spectral line profiles

## 1. Introduction

Novel X-ray spectral line shapes diagnostics in ultra-high intensity laser–plasmas have been constructed in recent years, uncovering a large amount information on the relativistic interactions characterizing the plasmas. This review is dedicated to the state-of-the-art of the improved spectroscopic methods adjusted for the analysis of the experimental results from Vulcan Petawatt (PW) laser facility at the Rutherford Appleton Laboratory (RAL) [1]. These methods are based on the intra-Stark spectroscopy (ISS) which is spectroscopy within the quasi-static Stark profile of a spectral line when radiating ions are subjected simultaneously to a quasi-static field $F$ and to a quasi-monochromatic electric field $E(t)$ at the characteristic frequency $\omega$. Due to non-linear phenomena, some dips occur at certain locations of the quasi-static Stark profile. These dips correspond to a non-linear dynamic resonance between the Stark splitting $\omega_F$ of the spectral line and the frequency $\omega$ of the electric field or its harmonics. Details on the theory of ISS can be found in [2–5].

In the plasmas of interest, the quasi-monochromatic electric field $E(t)$ represents a Langmuir wave. The corresponding structures in the spectral line profiles are the Langmuir-wave-induced

dips (L-dips) and they were first exhibited in the gas-liner pinch experiments performed by Kunze's group [4] providing a passive spectroscopic method for measuring with a high accuracy the electron density in plasmas and the amplitude of the Langmuir waves produced by nonlinear physical processes. Then experimental studies of L-dips were implemented in laser-produced plasmas obtained first with moderate laser intensities $(1 - 2) \times 10^{14}$ W/cm$^2$ [6–10]. It was shown that the ISS X-ray spectroscopy of these plasmas helped to reveal the physics of the laser–plasma interaction, leading to the generation of the Langmuir field. Later, the experiments were performed using a femtosecond laser driven cluster-based plasma, the laser intensity $(0.4 - 3) \times 10^{18}$ W/cm$^2$ being at the threshold for relativistic laser–plasma interaction [11]. In these experiments the L-dips observed were caused by Langmuir waves resulting from the two-plasmon decay instability. The present review is totally dedicated to experimental studies performed with incident laser intensities as high as $10^{20}$–$10^{21}$ Wcm$^{-2}$ and generating strongly relativistic laser–plasma interactions [12,13]. It advances the significant improvements developed at RAL, using the Optical Parametric Chirped Pulse Amplification technology (OPCPA) for the laser pulse generation and a plasma mirror for increasing the laser contrast and as a consequence diminishing the importance of the laser pre-plasma. High-resolution spectroscopy using a Focusing Spectrometer with Spatial Resolution (FSSR) allowed the analysis of SiXIV Ly-beta, Ly-gamma, and AlXIII Ly-beta lines. The reproducibility of the interpretation of spectroscopic results obtained from different shots of about the same laser intensity allowed the demonstration of the simultaneous production of the *Langmuir high frequency waves* and, for the first time, the *ion acoustic low frequency waves*, in high-density plasmas. Both kinds of waves have been predicted and attributed to the development of the non-linear physical process Parametric Decay Instability (PDI) developing at the surface of the *relativistic* critical density [13,14].

After an in-depth description of the experiments devoted to relativistic laser–plasmas at RAL in Section 2, the theory of ISS for those plasmas is reviewed in Section 3 including an overview of the theory of Langmuir-wave-caused dips for plasmas and two recent thorough theoretical studies: a robust computational method for fast calculations of line shapes affected by a low-frequency electrostatic plasma turbulence (i.e., the ion acoustic wave) and a rigorous analysis of the dynamic resonance condition involved in the ISS spectroscopy. The interpretation of the experimental profiles with the theoretical profiles is provided in Section 4. The support of the results with PIC simulations is detailed in Section 5. Finally, the conclusion (Section 6) enhances the first experimental discovery of the PDI non-linear process in relativistic laser–plasma interactions.

## 2. X-ray Spectra Measurements in Relativistic Laser–Plasma Interactions

The experiments were performed at Vulcan Petawatt (PW) facility at the Rutherford Appleton Laboratory [12,13], which provides a beam using optical Parametric, Chirped Pulse Amplification (OPCPA) technology with a central wavelength of 1054 nm and a pulse Full-Width-Half-Maximum (FWHM) duration, which could be up to 1500 fs. The OPCPA approach associated with a plasma mirror enables an amplified spontaneous emission to the peak-intensity contrast ratio exceeding $10^{-11}$ several nanoseconds before the peak of the laser pulse [15–17]. The laser pulse was focused with an f/3 off-axis parabola. At the best focus approximately 300 J on the target was contained within a 7 μm (FWHM) diameter spot providing a maximum intensity of $1.4 \times 10^{21}$ W/cm$^2$. The horizontally polarised laser beam was incident on target at 45° from the target surface normal, as shown schematically in Figure 1a.

The X-ray emission of the plasma was registered by means of a high luminosity focusing spectrometer, with a spatial resolution (FSSR) [18], at the directions close to the normal to the target surface, from the rear side of the target. To obtain the spectra with a high spectral resolution $(\lambda/\delta\lambda \sim 3000)$ in a rather broad wavelength window the FSSR was equipped with quartz/mica spherically bent crystals [19,20] settled to record K-shell emission of Rydberg H-like spectral lines of multi charged SiXIV and AlXIII ions. For reducing the level of noise caused by the background fogging and crystal fluorescence, a pair of 0.5 T neodymium–iron–boron permanent magnets (not reproduced in Figure 1a), which formed a slit of 10 μm wide, was placed in front of the crystal. The spectra were

recorded by means of Fujifilm TR Image Plate (IP) detectors protected against the exposure to the visible light by two layers of 1 μm-thick polypropylene $(C_3H_6)_n$ with 0.2 μm Al coating. Additionally, to prevent saturation of the detectors, the Mylar $(C_{10}H_8O_4)$ filter of 5 μm thickness was placed at the magnet entrance. Moreover, some of pure Si and Al targets were coated with or buried into thin plastic (CH) in order to keep the plasma of the main layer at solid densities regardless of the impact of the laser pre-pulse.

Figure 1b shows as examples the experimental spectra of Si XIV $Ly_\beta$ and $Ly_\gamma$ lines in two different shots with duration of 600 fs and laser intensities at the surface of the target theoretically estimated as $1.01 \times 10^{21}$ W/cm$^2$ and $0.24 \times 10^{21}$ W/cm$^2$, respectively. The positions of the dips/depressions in the profiles are marked in insets by vertical lines separated either by $2\lambda_{pe}$ or $4\lambda_{pe}$, where $\lambda_{pe} = [\lambda_0^2/(2\pi c)]\omega_{pe}$ ($\lambda_0$ is the unperturbed wavelength of the corresponding line and $\omega_{pe}$ the plasma electron frequency).

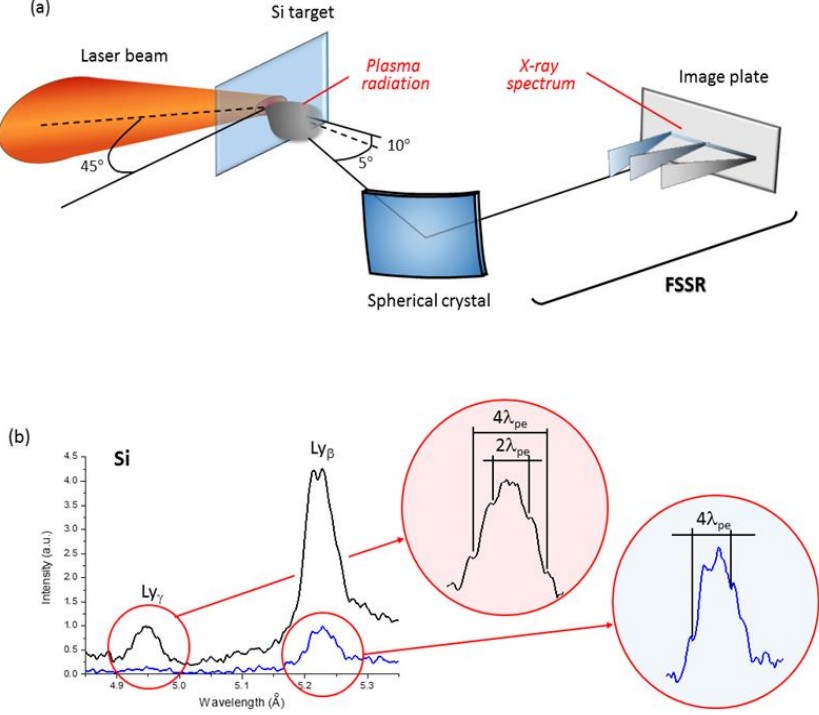

**Figure 1.** Schematic of experimental setup and typical X-ray spectra in the range of 0.485–0.535 nm. Experimental setup (**a**) and profiles (**b**) of Si XIV spectral lines, obtained in a single laser shot with initial laser intensity at the surface of the target estimated as $1.01 \times 10^{21}$ W/cm$^2$ (black trace) and $0.24 \times 10^{21}$ W/cm$^2$ (blue trace). In the insets, positions of the dips/depressions in the profiles are marked by vertical lines separated either by $2\lambda_{pe}$ or $4\lambda_{pe}$, where $\lambda_{pe} = [\lambda_0^2/(2\pi c)]\omega_{pe}$ ($\lambda_0$ is the unperturbed wavelength of the corresponding line and $\omega_{pe}$ the plasma electron frequency) [12].

## 3. Theory of Intra Stark Spectroscopy ISS for Relativistic Plasmas

### 3.1. The Langmuir-Wave-Caused Dips for Plasmas Dominated by a Low Frequency Electrostatic Turbulence LET

The Langmuir-wave-caused dips (L-dips) are the signatures of non-linear processes with generation of Langmuir high frequency waves. They result from a multifrequency resonance between the Stark splitting $\omega_F = 3n\hbar F/(2Z_r m_e e)$ of the energy levels (given here for hydrogen lines), caused by the quasi-static field $F$ in the plasma, and the frequency $\omega_L$ of the Langmuir waves, which practically coincides with the plasma electron frequency $\omega_{pe}(Ne)$: $\omega_F = s\omega_{pe}(N_e)$, $s = 1, 2, \ldots$ Here $n$ and $Z_r$ are the principal quantum number and the nuclear charge of the radiating atom/ion (radiator) respectively,

*Ne* stands for the electronic density and *s* for the number of quanta of the field involved in the resonance, $\hbar$ is the Plank constant, $e$ and $m_e$ are the electron charge and mass [2]. The quasi-static field $F$ represents the low-frequency part of the ion micro-field and the low Frequency Electrostatic Turbulence (LET), if the latter is developed in the plasma. This will be the case in relativistic plasmas as predicted by PIC simulations [12]. The physical mechanism producing simultaneously the Langmuir waves and the LET (specifically, the ion acoustic turbulence) out of the laser field is the Parametric Decay Instability (PDI). PDI is a nonlinear process, in which an electromagnetic wave decays into a Langmuir wave and an ion acoustic wave at the surface of the critical density $N_c$. It is important to remark that due to the broad distribution of the low frequency field $F$, there is always a fraction of radiators for which the resonance condition is satisfied. Even though the electric field of the Langmuir wave is considered to be monochromatic ($E_0\ cos\omega_{pe}\ t$), it produces a multi-frequency resonance as has been shown in paper [5].

The resonance condition $\omega_F = s\ \omega_{pe}(N_e)$, translates into specific locations of L-dips in spectral line profiles depending on $N_e$. In the following two specific cases are discussed for the Ly-lines.

In the case where the quasi-static field $F$ is dominated by the LET, the distance of an L-dip from the unperturbed wavelength $\lambda_0$ is given by $\Delta\lambda_{dip}(q, N_e) = -\left[\lambda_0^2/(2\pi c)\right]qs\omega_{pe}(N_e)$. Here $q = n_1 - n_2$ is the electric quantum number expressed via the parabolic quantum numbers $n_1$ and $n_2$: $q = 0, \pm 1, \pm 2, \dots$, $\pm (n-1)$. It labels Stark components of Ly-lines. For a pair of Stark components, corresponding to $q$ and $-q$, there could be multi-quantum resonances ($s = 1, 2 \dots$) leading to different pairs of L-dips located symmetrically in the red and blue parts of the profile: $\Delta\lambda_{dip}(N_e) = \pm\left[\lambda_0^2/(2\pi c)\right]qs\omega_{pe}(N_e)$.

If the quasi-static field $F$ is dominated by the ion micro-field, then the above formula for $\Delta\lambda_{dip}$ would hold only for relatively low electron densities. For relatively high electron densities, the spatial non-uniformity of the ion micro-field has to be taken into account leading to the result [2,21]:

$$\Delta\lambda_{dip}(q, N_e) = -\frac{\lambda_0^2}{2\pi c}\left[qs\omega_{pe} + \frac{2(s\omega_{pe})^3}{27n^3 Z_r Z_p \omega_{at}}\right]^{\frac{1}{2}}\left[n^2\left(n^2 - 6q^2 - 1\right) + 12n^2q^2\right] \tag{1}$$

where $Z_p$ is the charge of the perturbing ions, $Z_r$ the charge of the radiating ions and $\omega_{at} = m_e e^4/\hbar^3 = 4.14 \cdot 10^{16} s^{-1}$ the atomic unit of frequency. The first, primary term in braces reflects the dipole interaction with the ion micro-field. The second, smaller term in braces takes into account—via the quadrupole interaction—a spatial non-uniformity of the ion micro-field. It results in the shift of the midpoint between the pair of L-dips, corresponding to $q$ and $-q$, with respect to the unperturbed wavelength. Thus, there is a signature of the pre-dominance of the LET when the symmetrical location of the L-dips in the red and blue parts of the experimental profile is observed.

In any case (whether the field $F$ is dominated by the ion micro-field or by a LET), the separation of the two L-dips, corresponding to $q$ and $-q$, is:

$$\Delta\lambda_{dip}(N_e) = \pm\left[\lambda_0^2/(2\pi c)\right]q\omega_{pe}(N_e) \tag{2}$$

thus enabling measurement of $N_e$. It is important to emphasize that this passive spectroscopic method for measuring $N_e$ is just as accurate as the active spectroscopic method using the Thompson scattering, [4,21]. From the experimental ISS results analysis, the simultaneous signatures of the Langmuir waves and the LET confirmed that the PDI nonlinear process actually took place at the critical density $N_c$.

The half-width of the L-dip (i.e., the separation between the dip and the nearest "bump"), is controlled by the amplitude $E_0$ of the Langmuir wave [2]:

$$\delta\lambda_{1/2} \approx \left(\frac{3}{2}\right)^{\frac{1}{2}}\frac{\lambda_0^2 n^2 E_0}{8\pi m_e ec Z_r} \tag{3}$$

Thus, by measuring the experimental half-width of L-dips, one can determine the amplitude $E_0$ of the Langmuir wave.

Before analyzing the experimental spectral lines, we should note the following. The laser frequency $\omega$ corresponds to the wavelength $\lambda$ of approximately 1054 nm. At lower, non-relativistic laser intensities, the theoretical electron density $N_e$ determined from the equation $\omega = \omega_{pe}$, where $\omega_{pe} = (4\pi e^2 N_e / m_e)^{\frac{1}{2}}$, would be $1.0 \times 10^{21}$ cm$^{-3}$. However, at the laser intensities ~$10^{21}$ W/cm$^2$, i.e., those corresponding to the experiment, due to relativistic effects, the "relativistic" critical electron density $N_{cr}$ becomes higher than the critical density $N_c$ [22–24]. For the linear polarized laser radiation, it becomes [25]:

$$N_{cr} = \frac{\left(\frac{\pi a}{4}\right) m_e \omega^2}{4\pi e^2} , a = \lambda(\mu m)\left[\frac{I\left(\frac{W}{cm^2}\right)}{1.37 \times 10^{18}}\right]^{\frac{1}{2}} \tag{4}$$

The ability of the ultra-intense laser radiation to penetrate into regions of the density higher than $N_c$—i.e., a relativistically-induced transparency regime—has been previously demonstrated by PIC simulations [26]. In the present paper, we not only took into account the effect of the relativistically induced transparency (details of which can be found in paper [27]) on the resonance condition, but also confirmed that the L-dips originate from the region of the relativistic critical density (which is greater than the non-relativistic critical density), as shown in Section 4.

### 3.2. Rigorous Condition of the Dynamic Resonance

The resonance condition for the formation of the dips $\omega_{stark}(F_{res}) = s\omega_{pe}(N_e)$ with $\omega_{stark}(F_{res}) = 3n\hbar F_{res}/(2Z_r m_e e)$ (deeply studied for hydrogen lines in Section 3.1) is only valid for $E_0/F_{res} << 1$, this condition allowing the determination of the resonant value of the quasi-static field $F_{res}$ independent of $E_0$.

The rigorous resonant condition valid for any value of the ratio $E_0/F_{res}$ reads:

$$\omega_{stark}(F_{res})\, g(\varepsilon) = s\omega_{pe}\,(N_e), g(\varepsilon) = (1 + \varepsilon^2)^{1/2}\, EllipticE[\varepsilon/(1 + \varepsilon^2)^{1/2}], \varepsilon = E_o/F_{res}, \tag{5}$$

where $\omega_{stark}(F_{res}) = 3n\hbar F_{res}/(2Z_r m_e e)$ for hydrogen lines and *EllipticE*[ ... ] is the complete elliptic integral of the second kind [2] (pp. 49, 28; Appendix G). When the ratio $E_0/F_{res} > 0.5$, L-dips cannot exist and instead, in the profile, there could appear a sequence of satellites at the resonance frequencies [28] (Appendix L). It is important to emphasize that this resonance condition involving now both $E_0$ and $F_{res}$ (5) is relevant for some shots during the experimental campaigns, explaining the disappearance of the Langmuir dips.

Several notes should be made. First, in our experimental conditions, the low-frequency electric field in the plasma is dominated by the LET, rather than by the ion microfield. Second, the resonant value of the low-frequency electric field $F_{res}$, defined by Equation (5), is much greater than the characteristic value $F_{ion}$ of the ion microfield. Therefore, the field of the value $F_{res}$ can be considered quasi-static even if much smaller fields ~$F_{ion}$ would not be quasi-static. Third, we did not assume the entire ion microfield to be quasi-static.

### 3.3. Robust Computation of Line Shapes Affected by the LET

For a detailed quantitative analysis of the ISS results from relativistic laser–plasma, we calculated the theoretical profiles as follows. In this computation, the total quasi-static field $F$ is the vector sum of two contributions: $F = F_t + F_i$. The first contribution $F_t$ is the field of the LET, while the second contribution $F_i$ is the quasi-static part of the ion micro-field. We employed the results from papers [29–31] to calculate the distributions of the total quasi-static field $F = F_t + F_i$. Specifically, we calculated the distribution of the total quasi-static field $F$ in the form of the convolution of the APEX (Adjustable-Parameter EXponential approximation) distribution of $F_i$ [30] with the Rayleigh-type

distribution of $F_t$ [29]. We note that for the case where the characteristic value of the LET is much greater than the characteristic value of the ion micro-field, the analytical results from paper [31] provide a robust way to calculate the distribution of the total quasi-static field without calculating the convolution. We also took into account the broadening by the electron micro-field, by the dynamical part of the ion micro-field, the Doppler and the instrumental broadenings (~0.1 mÅ), as well as the theoretically expected asymmetry of the profiles [32,33]. For calculating the details of the spectral line shape in the regions of L-dips we employed the analytical solution from [5] for the wave functions of the quasi-energy states, the latter being caused simultaneously by all harmonics of the total electric field $E(t) = F + E_0 \cos(\omega_{pe}t)$ (vectors $F$ and $E_0$ are not collinear).

In the present paper, all theoretical profiles obtained with the above robust computations will be compared to the results performed using the code FLYCHK [34]. This code, that does not take into account the Stark broadening by the LET and the presence of the L-dips, leads, when comparing with the experimental results, to very high electronic densities 1–3 $10^{23}$ cm$^{-3}$ inconsistent with the relativistic critical electron density $N_{cr}$ = 2–3 $10^{22}$ cm$^{-3}$. The variation of the FLYCHK code used takes into account both Doppler and instruments broadenings.

## 4. Analysis of the Experimental Spectra and Comparison with Spectral Line Shapes Codes Simulations

### 4.1. First Discovery of the Ion Acoustic Turbulence in Relativistic Laser–Plasmas Using X-ray Spectroscopy

Let us consider the experimental profiles of Si XIV Ly$_\gamma$, produced in an interaction with a single laser pulse with initial laser intensity at the surface of the target estimated as $1.01 \times 10^{21}$ W/cm$^2$ (black trace in Figure 1b), and Si XIV Ly$_\beta$, produced in an interaction with a single laser pulse with initial laser intensity at the surface of the target estimated as $0.24 \times 10^{21}$ W/cm$^2$ (blue trace in Figure 1b) [12]. For these profiles the peak-intensity contrast ratio was $10^9$. We mention that L-dips were not observed in the experimental profile of Si XIV Ly$_\beta$ line in the black trace Figure 1b. It seems that a process of self-absorption of radiation prevented L-dips from being visible in this line. We note that without the self-absorption the Ly$_\beta$ line would show a relatively deep central minimum (related not to the the L-dips, but to the structure of the Stark splitting of this line) and shallow L-dips. The self-absorption completely washes out the L-dips but does not completely washes out the central minimum: it just makes it shallow instead of being relatively deep. As for the Ly$_\gamma$ line in the black trace Figure 1b, at the possible locations of the L-dips, the experimental profile already merged into the noise.

The experimental black trace profile Si XIV Ly$_\gamma$ shows a pair of two L-dips nearest to the line center ($q = \pm 1$, $s = 1$) separated from each other by 28 mÅ yielding an electron density of $N_e = 3.6 \times 10^{22}$ cm$^{-3}$. In addition to this, a pair of L-dips, separated from each other by 56 mÅ, were also observed, representing a superposition of two pairs of the L-dips: $q = \pm 2$, $s = 1$ and $q = \pm 1$, $s = 2$. From the Equation (2) this yields the same $N_e = 3.6 \times 10^{22}$ cm$^{-3}$, thus reinforcing the interpretation of the experimental dips as the L-dips—caused by the resonant interaction of the Langmuir waves, developed at the surface of the relativistic critical density, with the quasi-static electric field. Let us remark that at the initial laser intensity at the surface of the target estimated as $1.01 \times 10^{21}$ W/cm$^2$, the relativistic critical density would be $N_{cr} = 2.3 \times 10^{22}$ cm$^{-3}$ according to Equation (4). However, due to several physical effects [26,27] (self-focusing of the laser beam, Raman and Brillouin back scattering), the actual intensity of the transverse electromagnetic wave in the plasma can be significantly higher than the intensity of the incident laser radiation at the surface of the target. In the present experiment for the relativistic critical density to be approximately equal to the density $N_e = 3.6 \times 10^{22}$ cm$^{-3}$ deduced by the spectroscopic analysis, it would require the enhancement of the initially estimated intensity of the transverse electromagnetic wave at the surface of the target due to the above physical effects just by a factor of two. In both cases, L-dips separated by 28 mÅ and for the L-dips separated by 56 mÅ, the mid-point between the two dips in the pair practically coincides with the unperturbed wavelength $\lambda_0$. This is a strong indication that the quasi-static field $F$ was dominated by the LET. If the LET would be absent, then according to Equation (1), the mid-point

of the pair of the L-dips separated by 28 mÅ should have been shifted by 5.8 mÅ to the red with respect to $\lambda_0$ and the mid-point of the pair of the L-dips separated by 56 mÅ would have been similarly shifted by 10.7 mÅ to the red with respect to $\lambda_0$, these shifts being due to the spatial non-uniformity of the ion micro-field reflected by the second term in Equation (1). Another strong indication of the presence of the LET observed from the above result comes from the analysis of the broadening of this spectral line that was performed using code FLYCHK [34]. This code, which does not take into account the Stark broadening by the LET (and the presence of L-dips), yielded $N_e = 0.9 \times 10^{23}$ cm$^{-3}$, i.e., almost three times higher than the actual $N_e = 3.6 \times 10^{22}$ cm$^{-3}$ for the best fit to the experimental profile as shown in Figure 2a. This result is not reasonable.

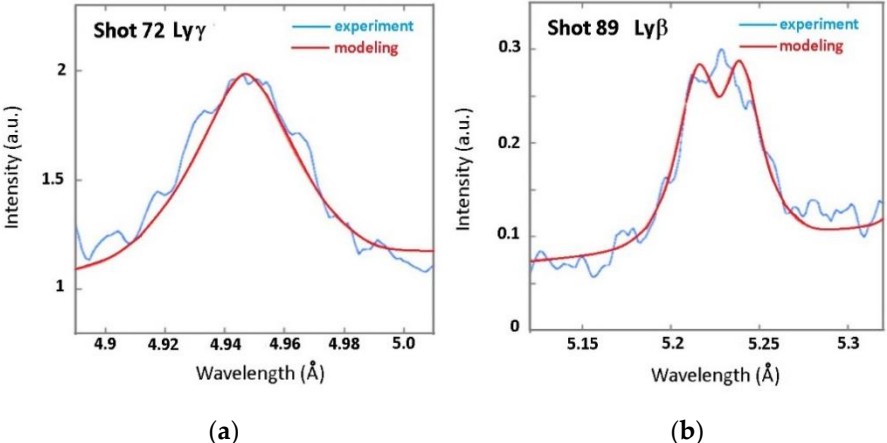

**Figure 2.** Experimental spectra and their comparisons with FLYCHK modeling. (**a**) Comparison of the experimental profile of Si XIV Ly$\gamma$ line in shot with initial laser intensity at the surface of the target estimated as $1.01 \times 10^{21}$ W/cm$^2$ with a simulation performed using a variation of the code FLYCHK for calculating the Stark broadening, then adding both Doppler and instrument broadening, and if necessary, opacity. The L-dips phenomenon and the spectral line broadening by LET were not included in the FLYCHK. The plasma parameters for the best fit are $N_e = 0.9 \times 10^{23}$ cm$^{-3}$ and $T_e = 500$ eV. (**b**) The same as in (**a**) but for Si XIV Ly$\beta$ line produced in a single laser shot with initial laser intensity at the surface of the target estimated as $0.24 \times 10^{21}$ W/cm$^2$; $N_e = 3 \times 10^{23}$ cm$^{-3}$ and $T_e = 500$ eV [12].

The analysis of the experimental profile of Si XIV Ly$_\beta$ with the initial laser intensity at the surface of the target estimated as $0.24 \times 10^{21}$ W/cm$^2$ (blue trace in Figure 1b), shows a situation similar to Si XIV Ly$_\gamma$. There is a pair of the L-dips separated from each other by 43 mA. The electron density deduced from the separation within the pair of these L-dips, is $N_e = 1.74 \times 10^{22}$ cm$^{-3}$ (assuming $|q|s = 2$ in Equation (2)). This pair of dips corresponds either to the Stark component $q = 1$ for a two-quantum resonance $s = 2$ or to the Stark component $q = 2$ for the one-quantum resonance $s = 1$. The superposition of two different dips at the same location results in a L-super-dip with a significantly enhanced visibility. The location of the would-be L-dips, corresponding to $|q|s = 1$, is too close to the central, most intense part of the experimental line profile so that they are not observed either due to a self-absorption in the most intense part of the profile, or because the relatively small values of the field $F$, corresponding to the central part of the profile, are not quasi-static. The mid-point between the two dips in the pair practically coincides with the unperturbed wavelength $\lambda_0$. This is again a strong indication that the quasi-static field $F$ was dominated by the LET. If the LET would be absent, then according to Equation (1), the mid-point of this pair of L-dips should have been shifted by 6.1 mÅ to the red with respect to $\lambda_0$. Another strong indication of the presence of the LET in the profile of Si XIV Ly$_\beta$ comes from the analysis performed using code FLYCHK [34] (Figure 2b). An electron density of $N_e = 3 \times 10^{23}$ cm$^{-3}$, was obtained, i.e., 17 times higher than the experimentally verified $N_e = 1.74 \times 10^{22}$ cm$^{-3}$. This is yet another strong indication of an additional Stark broadening

by the LET (not accounted for by FLYCHK). At the initial laser intensity at the surface of the target estimated as $0.24 \times 10^{21}$ W/cm$^2$, the relativistic critical density would be $N_{cr} = 1.1 \times 10^{22}$ cm$^{-3}$ according to Equation (4). However, again due to the self-focusing, as well as Raman and Brillouin backscattering, the actual intensity of the transverse electromagnetic wave in the plasma can be significantly higher. In the present case, for the relativistic critical density to be approximately equal to the density $N_e = 1.74 \times 10^{22}$ cm$^{-3}$ deduced by the spectroscopic analysis, again it would require the enhancement of the intensity of the transverse electromagnetic wave due to the above physical effects just by a factor of two.

From both analysis of the Si XIV Ly$_\gamma$ and Si XIV Ly$_\beta$ profiles, it can be confirmed that the LET developed simultaneously with the Langmuir waves at the relativistic critical density surface and thus it is most likely to be an ion acoustic turbulence. The most probable and the best studied mechanism for developing Langmuir waves at the surface of the relativistic critical density is a parametric decay, which is a nonlinear process where the pump wave ($t_1$) excites both the Langmuir wave ($l$) and an ion-acoustic wave ($s$): $t_1 \rightarrow l + s$.

### 4.2. Robust Computations for the Analysis of the Experimental Si XIV Ly$_\gamma$ and Si XIV Ly$_\beta$ Profiles

The robust computations including L-dips and the spectral line broadening by LET (Section 3.3) performed for the analysis of the Si XIV Ly$_\gamma$ and Si XIV Ly$_\beta$ profiles are given in Figure 3a,b. The comparisons with the experimental spectra is fruitful as confirmed in the following.

From the experimental profile of Si XIV Ly$_\gamma$ (Figure 3a), it follows that the root-mean-square value of $F_t$ was $F_{t,rms} = 2.1$ GV/cm. For comparison, the characteristic ion microfield $F_{i,typ} = 2.603\ eZ^{1/3}N_e^{2/3}$ was 1.0 GV/cm. From the half width of the experimental L-dips, by using Equation (3), we found the amplitude of the Langmuir wave to be $E_0 = 0.6$ GV/cm. The resonant value of the quasi-static field $F_{res}$, determined by the condition of the resonance between the separation of the Stark sublevels and the plasma frequency $3n\hbar F_{res}/(2Z_r m_e e) = \omega_{pe}$, was 3.1 GV/cm, so that the validity condition for the existence of L-dips $E_0 < F_{res}$ (Section 3.2) was satisfied.

From the experimental profile of Si XIV Ly$_\beta$ (Figure 3b), it follows that the root-mean-square value of $F_t$ was $F_{t,rms} = 3.9$ GV/cm. For comparison, the characteristic ion microfield was $F_{i,typ} = 0.6$ GV/cm. From the halfwidth of the experimental L-dips, by using Equation (3), we found the amplitude of the Langmuir wave to be $E_0 = 1.0$ GV/cm. The resonant value of the quasi-static field was $F_{res} = 5.8$ GV/cm, so that the validity condition for the existence of L-dips $E_0 < F_{res}$ was again satisfied.

The good agreement between experimental spectra and simulations reinforces the discovery of the simultaneous production of LET with the Langmuir waves. We note that the electron densities involved turned out to be much lower than the densities deduced using FLYCHK simulations, which ignored the LET and the L-dips. The densities used for the robust computations are in perfect agreement with the densities deduced experimentally from the dips separations.

We also performed experiments where the spectrometer viewed the laser-irradiated front surface of the target. As an example, Figure 4 shows the experimental spectrum of Al XIII Ly$_\beta$ line (4 μm Al foil coated by 0.45 μm CH), which was obtained in a single laser shot with duration of 0.9 ps and the laser intensity at the surface of the target theoretically estimated as $6.7 \times 10^{20}$ W/cm$^2$. The peak-intensity contrast ratio was $10^9$ as for the experimental spectra Si XIV Ly$_\beta$ and Ly$_\gamma$. This spectrum exhibits two pairs of L-dips: one pair—at $\pm$ 16.8 mÅ from the line center, another pair—at $\pm$ 33.6 mÅ from the line center. The two dip pairs (two pairs of dips) are symmetrical with respect to the line center, confirming the production of LET with the Langmuir waves. Also, the density used for the robust computation is in perfect agreement with the density deduced experimentally from the dips separations.

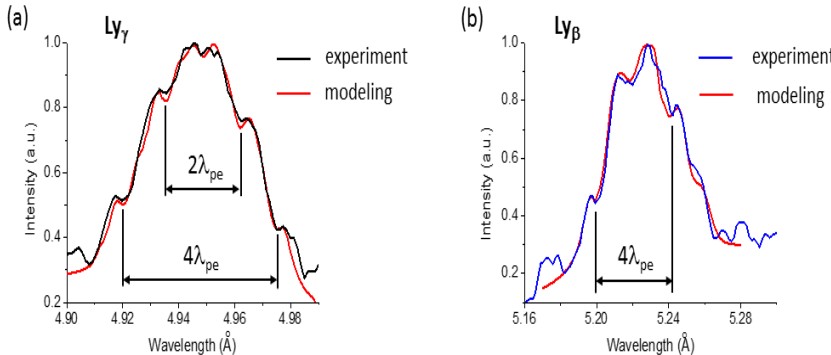

**Figure 3.** Experimental spectra and their comparisons with a different Stark broadening code including L-dips and the spectral line broadening by LET. (**a**) Experimental spectra of Si XIV $Ly_\gamma$ line in a single laser shot with initial laser intensity at the surface of the target estimated as $1.01 \times 10^{21}$ W/cm$^2$ initial laser intensity at the surface of the target. The positions of the dips/depressions in the profiles are marked by vertical lines separated either by $2\lambda_{pe}$ or $4\lambda_{pe}$, where $\lambda_{pe} = [\lambda_0^2/(2\pi c)]\omega_{pe}$ ($\lambda_0$ is the unperturbed wavelength of the corresponding line). Also shown is a theoretical profile at $N_e = 3.6 \times 10^{22}$ cm$^{-3}$ allowing, in particular, for a low-frequency electrostatic turbulence. (**b**) Same but for Si XIV $Ly_\beta$ line in a single laser shot with initial laser intensity at the surface of the target estimated as $0.24 \times 10^{21}$ W/cm$^2$ initial laser intensity at the surface of the target. The theoretical profile is shown at $N_e = 1.74 \times 10^{22}$ cm$^{-3}$ [12].

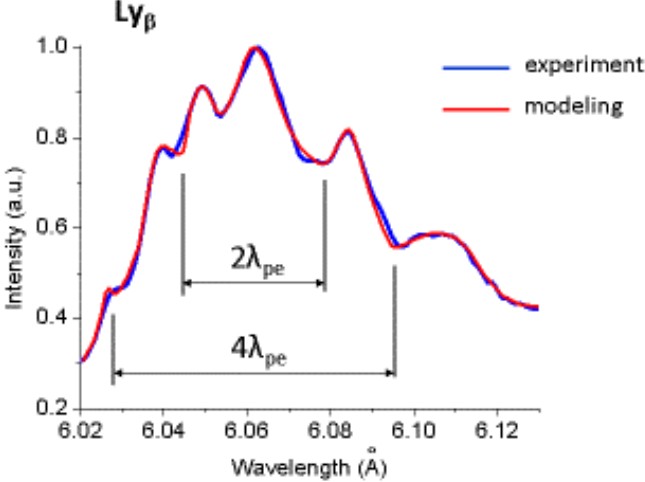

**Figure 4.** Experimental spectrum of Al XIII $Ly_\beta$ and its comparison with the Stark broadening code including L-dips and the spectral line broadening by LET. The experimental spectrum was obtained in a single laser shot with initial laser intensity at the surface of the target estimated as $6.7 \times 10^{20}$ W/cm$^2$. The positions of the L-dips in the profiles (the L-dips being very pronounced) are marked by vertical lines separated either by $2\lambda_{pe}$ or $4\lambda_{pe}$, where $\lambda_{pe} = [\lambda_0^2/(2\pi c)]\omega_{pe}$ ($\lambda_0$ is the unperturbed wavelength of the corresponding line). Also shown is a theoretical profile at $N_e = 2.35 \times 10^{22}$ cm$^{-3}$ allowing, in particular, for L-dips and for a low-frequency electrostatic turbulence. This value of Ne is much lower than Ne obtained from fitting the same experimental spectrum by code FLYCHK that did not include the L-dips phenomenon and the spectral line broadening by LET, and therefore had significant discrepancies with the experimental profile at the locations of the L-dips [12].

### 4.3. In-Depth Study of ISS in the X-ray Range Emission from Relativistic Laser–Plasmas

Recently new experiments have been performed at RAL [17] allowing a peak-intensity contrast ratio exceeding $10^{-11}$ (instead of $10^{-9}$ for the experimental results discussed in the paper up to now) due to a plasma mirror allowing an amplified spontaneous emission. An in-depth study was then

undertaken for the case Si XIV Ly$_\beta$ allowing a reliable reproducibility of the Langmuir-wave-induced dips at the same locations in the experimental profiles as well as of the deduced parameters (fields) of the Langmuir waves and ion acoustic turbulence in several individual 1 ps laser pulses and of the peak irradiance of 1 to $3 \times 10^{20}$ W/cm$^2$.

Figure 5 shows the comparisons of the robust computations, allowing in particular, for the LET and L-dips, with the corresponding experimental profiles from shots A, B, C. The L-super-dip is observed twice in the experimental profiles: one in the blue part and the other in the red part. These L-super-dips are located practically symmetrically at the distance $\Delta\lambda_{dip}(N_e) = 24$ mÅ from the unperturbed wavelength. According to Equation (3) with $|q|s = 2$, this translates into the electron density $N_e = 2.2 \times 10^{22}$ cm$^{-3}$ The theoretical profiles were calculated for this density $2.2 \times 10^{22}$ cm$^{-3}$ and the temperature $T = 600, 550,$ and $600$ eV for shots A, B, and C, respectively. The comparison demonstrates a good agreement between the theoretical and experimental profiles, and thus reinforces the good interpretation of these experimental profiles. Moreover, it reinforces that it was the PDI at the surface of the relativistic critical density that produced simultaneously the Langmuir waves and the ion acoustic turbulence in shots A, B, and C. The modeling of the experimental profiles A, B, C using the code FLYCHK [34] (Figure 6) confirmed once more the need to introduce the LET for the interpretation. This innovative code yielded $T = 500$ eV and $N_e = 6 \times 10^{23}$ cm$^{-3}$. This value of $N_e$ is one and a half orders of magnitude higher than the electron density $N_e = 2.2 \times 10^{22}$ cm$^{-3}$ deduced from the experimental L-dips.

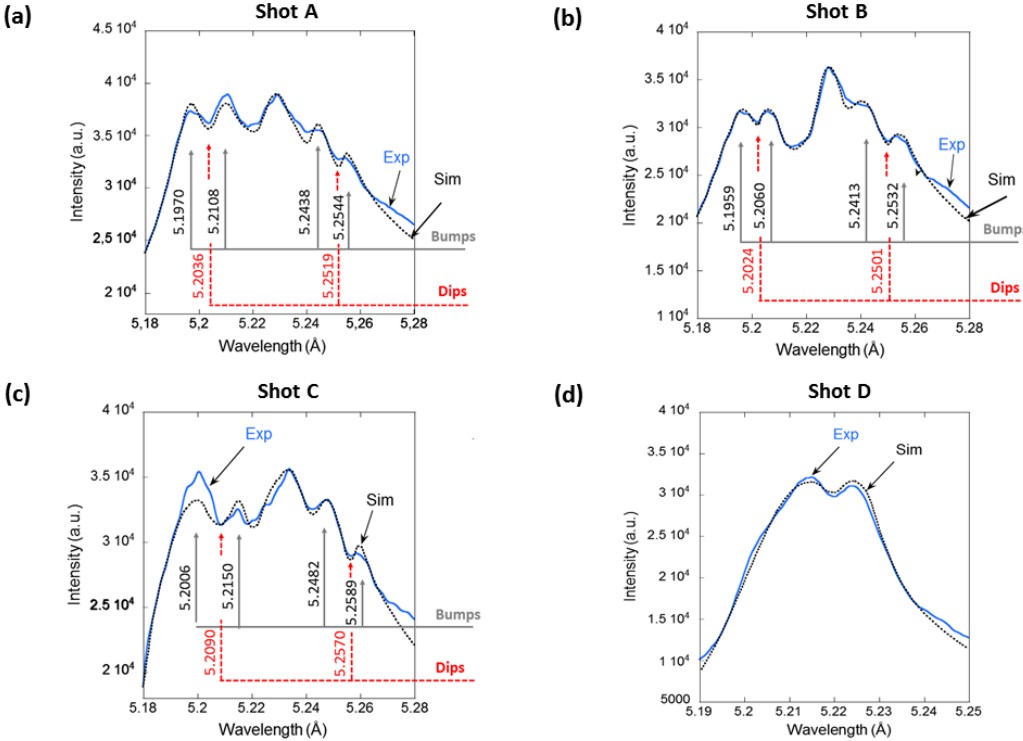

**Figure 5.** Comparison of the experimental profiles of the Si XIV Ly-beta line (solid line, blue in the online version, marked Exp) with the theoretical profiles (dotted line, black, marked Sim) allowing for the effects of the Langmuir waves, the LET, and all other broadening mechanisms (see the text). In the profiles from shots A, B, and C, there are clearly seen 'bump–dip–bump' structures (both in the red and blue parts of the profiles) typical for the L-dips phenomenon. The L-dips are not observed in shot D. The following parameters provided the best fit: (**a**) $N_e = 2.2 \times 10^{22}$ cm$^{-3}$, $T = 600$ eV, $F_{t,rms} = 4.8$ GV/cm, $E_0 = 0.7$ GV/cm; (**b**) $N_e = 2.2 \times 10^{22}$ cm$^{-3}$, $T = 550$ eV, $F_{t,rms} = 4.4$ GV/cm, $E_0 = 0.5$ GV/cm; (**c**) $N_e = 2.2 \times 10^{22}$ cm$^{-3}$, $T = 600$ eV, $F_{t,rms} = 4.9$ GV/cm, $E_0 = 0.6$ GV/cm; (**d**) $N_e = 6.6 \times 10^{21}$ cm$^{-3}$, $T = 550$ eV, $F_{t,rms} = 2.0$ GV/cm, $E_0 = 2.0$ GV/cm [13].

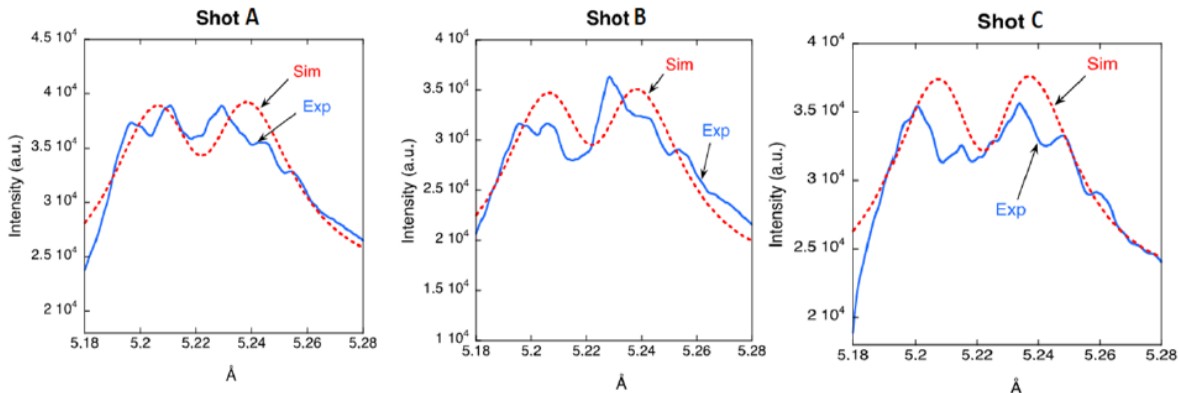

**Figure 6.** Experimental profiles of the Si XIV Ly-beta line in shots A, B, C (blue color in the online version) and their comparison with simulations using code FLYCHK (red color in the online version) at the electron density $N_e = 6 \times 10^{23}$ cm$^{-3}$ and the temperature $T = 500$ eV [13].

The determination of the fields of the Langmuir waves and ion acoustic turbulence can be deduced from the experimental Si XIV Ly$_\beta$ profiles A, B, C (Figure 5) and their robust computations. The values of the amplitude of the Langmuir waves using (3) have been obtained: $E_0 = 0.7, 0.5,$ and 0.6 GV/cm for shots A, B, and C, respectively. The resonant value of the quasi-static field $F_{res}$ responsible for the formation of the L-dips, can be determined from the resonance condition $\omega_F = s\, \omega_{pe}(N_e)$. For shots A, B, and C, it yields: $F_{res} = 6.5$ GV/cm for $s = 2$ and $F_{res} = 3.25$ GV/cm for $s = 1$. These values of $F_{res}$ are about 10 and 5 times higher than the Langmuir wave amplitude $E_0$, respectively. Thus, the condition $E_0 \ll F_{res}$, necessary for the formation of the L-dips, was fulfilled. These values for $F_{res}$ are coherent with the root-mean-square field values of the LET introduced for the robust computations: $F_{t,rms} = 4.8, 4.4,$ and 4.9 GV/cm for shots A, B, and C respectively. For comparison, the characteristic ion micro-field $F_{i,typ} = 2.603$ eZ1/3Ne2/3 was 1.5 GV/cm.

Now we proceed to analyze the experimental profile of the Si XIV Ly$_\beta$ line in shot D (Figure 5). The experimental profile does not show bump–dip–bump structures—in distinction to shots A, B, and C. In shot D, the incident laser intensity was $I = 8.8 \times 10^{19}$ W/cm$^2$, significantly lower than in shots A, B, and C. The corresponding relativistic critical density is $N_{cr} = 6.6 \times 10^{21}$ cm$^{-3}$. The modeling of this spectra using the code FLYCHK yielded $N_e = 1.7 \times 10^{23}$ cm$^{-3}$, which is one and a half order of magnitude higher than the relativistic critical density and by an order of magnitude higher than the region of the electron density $N_e = 2.2 \times 10^{22}$ cm$^{-3}$ from which the experimental profiles were emitted in shots A, B, and C is not reliable for the interpretation of shot D profile. The most probable interpretation of the experimental profile in shot D is the following.

In shot D the electron density was significantly lower than in shots A, B, and C. Therefore, the damping of the Langmuir waves was significantly lower, which could allow the Langmuir waves to reach a significantly higher amplitude. Figure 5d shows the comparison of the experimental profile from shot D with robust computing based on the code allowing for the LET and the Langmuir waves at $N_e = 6.6 \times 10^{21}$ cm$^{-3}$, $T = 550$ eV, $F_{t,rms} = 2.0$ GV/cm, $E_0 = 2.0$ GV/cm. It is seen that this theoretical profile is in a good agreement with the experimental profile and it does not exhibit bump–dip–bump structures. For high Langmuir wave amplitudes—i.e., when the ratio $E_0/F_{res} > 0.5$—the L-dips cannot form (see Section 3.3) and the resonant value of the quasi-static field is given by (5). For $N_e = 6.6 \times 10^{21}$ cm$^{-3}$ and $E_0 = 2.0$ GV/cm, Equation (5) yields $F_{res} = 1.7$ GV/cm for $s = 1$ (so that $E_0/F_{res} = 1.2$) and $F_{res} = 3.5$ GV/cm for $s = 2$ (so that $E_0/F_{res} = 0.6$). Thus, both for the one-quantum resonance ($s = 1$) and for the two-quantum resonance ($s = 2$), we get $E_0/F_{res} > 0.5$, so that the L-dips were not able to form. We note that, while for shots A, B, and C the electron density could be deduced from the locations of the L-dips and from the robust computation, in shot D the only one possibility is by modeling the entire experimental profile using the code that allows for the interplay of the LET and the Langmuir waves. Up to now, it seems to be the first experimental profile diagnosed in

relativistic laser–plasma interactions and explained with the simultaneous production of Langmuir waves and ion acoustic turbulence, but with no dips exhibited.

## 5. PIC Simulations for Relativistic Laser–Plasma Interactions

The results of this spectroscopic analysis have been also supported by PIC simulations. The simulations were performed using the modified 1D PIC code LPIC, in which a laser pulse of duration $t_L = 0.6$ ps and intensity $I_L = 2.7 \times 10^{21}$ W/cm$^2$ interacted with a Si target [35]. Laser pulse propagates along the $x$-axis and interacts with Si$^{+14}$ inhomogeneous plasma layer with a linear density ramp over the length L = 6 μm and a target thickness of 2 μm, with a constant ion density of $N_i = 6 \times 10^{22}$ cm$^{-3}$. Angle of incidence was set up 45° at P polarization. The results of the simulations are shown in Figure 7.

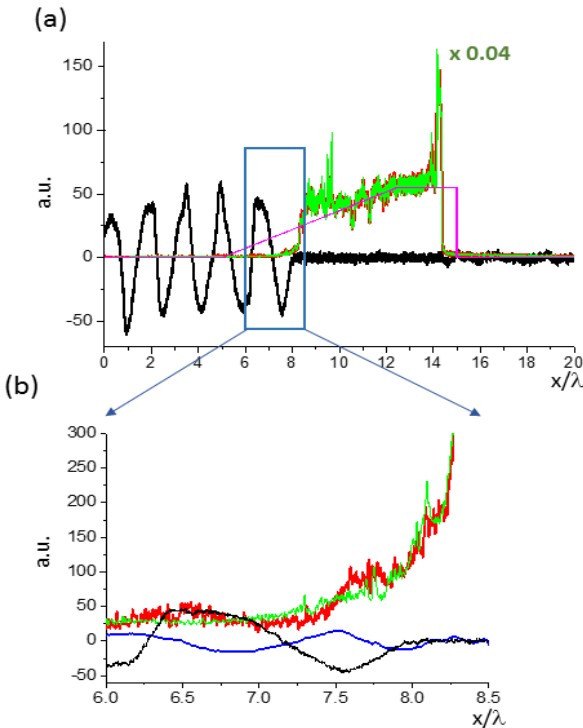

**Figure 7.** PIC simulations of the interaction of a 600 fs pulse of the intense (2.7 × 10$^{21}$ W/cm$^2$) linearly-polarized laser radiation with the Si$^{14}$ plasma layer. The general view of the laser-target interaction at the instant $t = 320$ fs from the beginning of the interaction (**a**) and the zoom (**b**) on the area shown by the blue rectangle in (**a**). The coordinate x is in units of the laser wavelength $\lambda$. The initial scaled density profile is presented by the violet line. The scaled electron and ion densities are presented by the red and green lines. They are given in the units of 0.04$ZN_{cr}$ in (**a**) and in the units of $ZN_{cr}$ in (**b**); here $N_{cr} = \gamma N_c$, where $N_c$ is the critical plasma density and $\gamma$ is the relativistic factor. The calculated scaled transversal electric field $a_L = eE_L/(m_e\omega_L c)$ is shown by the black line and the longitudinal one (the Langmuir wave) $a_l = eE_l/(m_e\omega_L c)$—by the blue line [12].

At the peak of the laser pulse intensity ($t = 320$ fs) it is seen that the scale of laser field decay is close to the scale of the plasma inhomogeneity. The longitudinal field of the Langmuir wave $E_l$ appeared in the vicinity of the point where the density is about one-quarter of the relativistic critical density. The Langmuir wave exists up to a point slightly above the relativistic critical density $N_{cr} = 3.6 \times 10^{22}$ cm$^{-3}$. In Figure 7b, the region near the relativistic critical density is $6\lambda < x < 8.25\lambda$: the small modulation of ion density in this region shown by the green line is the manifestation of the ion acoustic wave. Similar processes of such parametric decays ($t \rightarrow l + l', s$) were studied in the past [36], but for significantly lower laser intensities.

## 6. Conclusions

The evolution of the Intra-Stark X-ray Spectroscopy (ISS) in relativistic laser–plasma interactions during the last two years is presented here with a recent in-depth study of the simultaneous productions of Langmuir waves and of the ion acoustic turbulence at the surface of the relativistic critical density. We demonstrated, by improving the experiment at RAL, a reliable reproducibility of the spectroscopic signatures, i.e., the Langmuir dips of the Parametric Decay Instability (PDI) predicted by PIC simulations. The robust computations confirmed in details the experimental profiles, yielded the field parameters (Langmuir waves and ion acoustic turbulence), and demonstrated that the PDI process occurs at the relativistic critical density which is confirmed by the separation of the L-dips. The experiments revealed for the first time the situation where the PDI process could not lead to the appearance of L-dips when the Langmuir wave field is much larger than the LET field.

This study expands the Intra-Stark Spectroscopy (ISS) to relativistic laser–plasma situations and shows that the standard spectral line shape codes as FLYCHK are not adapted to reproduce these situations. The results presented in this review can be used for a better understanding of intense laser–plasma interactions and for laboratory modelling of physical processes in astrophysical objects.

**Author Contributions:** All authors contributed equally to this review.

**Funding:** This research received no external fundings.

**Acknowledgments:** The experimental results were obtained within the state assignment of FASO of Russia to JIHT, RAS (topic #01201357846).

**Conflicts of Interest:** The authors declare no conflicts of interest.

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
