# Peer review of "Mini-Review of Intra-Stark X-ray Spectroscopy of Relativistic Laser–Plasma Interactions"

_atoms, doi:10.3390/atoms6030045_

Round 1

Reviewer 1 Report

This is a paper in a very exciting field with interesting experimental results which should be published. There are, however, issues with  the theory and  missing important information from the manuscript, as well as  some claims on the theory that  ignore  the vast body of work by other colleagues and are detrimental to the work done.

1-2. Inroduction and Measurements

The presentation is clear.  The only thing missing in my opinion is
a clear  description of the target, i.e. Fig.1 (a) mentions a 'Si target'. If this is a pure Si target, it should be specified in the text. The reason for this insistsnce is that it is possibly important for the analysis, i.e. are the ion perturbing species Si ions or possibly other species which would be the case if one had a target like a Silicon oxide. Both    Fig. 1 and the PIC section seem to imply that  the first  assumption is true and I operate under this assumption in this report; however, I believe it would be clearer to the reader if the authors spelled it out in Section 2.

3. Theory of intra Stark spectroscopy

A monchromatic laser field in general may have a  number of effects
on spectral lines, e.g. satellites or  line narrowing.  That said,  the  L-dips theory is far from  accepted generally and I discuss this briefly in the Conclusions.
On a first reading, my reaction was the following: The authors  observed H-like Al and Si. This to me means quite high temperatures  if any substantial H-like emission is to be observed and this is confirmed by my rough calculations and the temperatures stated by the authors (see aso comment below). Then I would expect that a) Doppler broadening would dominate Stark  or at least compete and  b) the picture of static ions would be  inappropriate for say, Ly-gamma, if not even Ly-beta. Indeed, my calculations for L-gamma for Si  do not support the assumption of static ions. There may be valid reasons why ions are colder  than electrons (and such a treatment might be appropriate),  but the authors make no such discussion and use a single temperature value.
 In either case, the assumptions of the 'intra Stark' theory appear suspect.   I did some calculations and I am not at all sure the dips, if present at all, would not be masked by Doppler, which adds a significant broadening to say L-gamma.

I would also add that instead of  declaring the L-dips at least as reliable as Thomson scattering, a much more convincing step  would be to produce independent  plasma parameter measurements, and Thomson scattering could produce such an independent measurement. Indeed, if the L-dip theory has any merit, I  thnk the best way to demostrate it is to use  it on a well-diagnosed plasma by reliabe independent methods.

Also,  the last sentence of  the first paragraph of page 4 needs rewriting. Something like: "Even though
 the electric field of Langmuir waves is considered monochromatic, it still produces multi-frequency resonances[5]" is probably better english.

4. Analysis of the experimental spectra.

1)While the manuscript mentions inclusion of instrumental broadening in the analysis,  I do not see an actual figure or estimate for it.
This makes it very hard to judge  its relative importance. Same for opacity. The manuscript mentions that (Fig.2)
"with a simulation performed using a variation of the code FLYCHK for calculating the Stark broadening, then adding both Doppler and instrument broadening, and, IF necessary, opacity.", but I cannot understand what the analysis actually DID. So was it necessary to add opacity? How was this calculated? What effects did it have? And if opacity was necessary  in the FLYCHK calculations, was it also incuded in the L-dips calculation?  If so, how was it computed and if not, what was the reasoning apart from the  fact that  L-dips predicts lower densities? Was there any attempt to check for opacity experimentally? If not, can the authors elaborate on the (very understandable) experimental difficulties for doing so.
2)The authors claim that  "It seems that a process of self-absorption of radiation prevented L-dips from being visible
 in this line".  As I understand it, self-absorption would have been stronger in the line center, where the characteristic Ly-beta dip is seen and not obscured by self-absorption.  So it is very unclear to me why the dips, located significantly further from the line center would be distprted by self-absorption and the center not.
I would also like the authors to clarify if "It seems" is based on other physical considerations apart from "based on disagreement with the theoretical predictions".

3)  The authors mention acounting for asymmetries, but it is no clear to me what they did. Did they include fine structure? Did they account for nonquenching collisions and include states from n=5 for Ly-gamma?  Judging from  the blue Ly-beta line in Fig.1 , with a strong asymmetry, this would probably indicate strong gradients to me, which is not unexpected or unusual in similar experiments.

5. PIC simulations

Fig.7 mentions  that (b) is a zoom in on the area shown by the BLACK rectangle. Unless I am missing something, the author mean the BLUE rectange, right?

6. Conclusion

I do not really understand the fuss about FLYCHK being wrong and so on: FLYCHK is a great, SIMPLE, very easy to use code whose focus is NOT
  line profiles, especially not line profiles under extreme conditions, as is very clear -see for instance its home page.  There are other codes which  do include all the physics (and , may I say, much more than the L-dips, so I doubt the word 'advanced' is appropriate) and they are discussed in the SLSP workshops. For instance  the ion field need not be quasistatic. I believe the authors have done some very worthwhile work  that  should be published and need not   undermine it  by such claims.

Summing up, this is  a very interesting experimental work, which needs to  clarify the points raised above and
 adjust its focus on what the study did instead of extolling an unproven and in the view of most colleagues problematic theory.
The authors shoule use the L-dip theory or any other as they see fit, and note agreement or not with their work. That's fine. Calling it 'advanced' is hardly justified,   highly unjust to the work of many colleagues whose work includes a ot more physics  and is not helpful to their own work.

Author Response

Please see the attached file "Response to referee 1".

Reviewer 2 Report

this paper is acceptable for publication.

Author Response

Thank you very much for your absolutely positive review.